# Dietary and Nutritional Profiles among Brazilian Adolescents

**DOI:** 10.3390/nu14204233

**Published:** 2022-10-11

**Authors:** Diôgo Vale, Clélia de Oliveira Lyra, Natalie Marinho Dantas, Maria Eduarda da Costa Andrade, Angelo Giuseppe Roncalli da Costa Oliveira

**Affiliations:** 1Postgraduate Program in Collective Health, Federal University of Rio Grande do Norte, Natal 59056-000, RN, Brazil; 2Federal Institute of Education, Science and Technology of Rio Grande do Norte, Natal 59015-300, RN, Brazil; 3Postgraduate Program in Nutrition in Public Health, School of Public Health, University of São Paulo, São Paulo 05508-000, SP, Brazil

**Keywords:** adolescent, obesity, thinness, diet, social determinants of health

## Abstract

(1) Background: The present study analyzed the prevalence of dietary and nutritional profiles among Brazilian adolescents and their associations with social determinants of health. (2) Methods: A population-based survey was administered to 16,409 adolescents assessed by the 2015 National School Health Survey. A multivariate model of dietary and nutritional profiles was estimated from correspondence analysis. (3) Results: The dietary and nutritional profiles more prevalent among Brazilian adolescents were “lower nutritional risk dietary pattern and eutrophic” (42.6%), “lower nutritional risk dietary pattern and overweight” (6.8%), and “higher nutritional risk dietary pattern and overweight” (6.0%). Healthier profiles were associated with less urbanized territories, health-promoting behaviors, and families with worse material circumstances. The less healthy profiles were associated with more urbanized environments, health risk behaviors, and families with better material circumstances. (4) Brazilian adolescents have different dietary and nutritional profiles that are characterized by sociopolitical and economic contexts, family material and school circumstances, and the behavioral and psychosocial health factors of the individuals. All of this points to the social determination of these health problems among adolescents in Brazil.

## 1. Introduction

The prioritization of adolescents as a group for health monitoring is a global tendency because the transitions and experiences of this age group are associated with possible immediate or future health risks [1,2]. Thus, diet and nutrition are especially important because they are associated with behavioral and metabolic risk factors in the development of noncommunicable diseases [3].

Estimates of the adolescents’ nutritional status and dietary patterns are the usual methods in surveys designed with the objective of contributing to dietary and nutritional surveillance [4,5,6]. However, in this age group, some difficulties of associating nutritional status and dietary patterns are common in cross-sectional studies when assuming a causality hypothesis of adolescents with obesity having poorer dietary patterns [7].

Nutritional disorders (thinness, overweight, and obesity) are prevalent among Brazilian adolescent males [8]. The highest nutritional risk eating patterns are characterized by the more frequent consumption of ultra-processed food products and the less frequent consumption of fresh or minimally processed foods, which is prevalent among female adolescents [9]. This gender-related dichotomy suggests a significant effect observed in studies conducted to assess this population in Brazil including the Brazilian National School Health Survey (PeNSE) and the Cardiovascular Risk Study in Adolescents (ERICA). These studies did not identify direct associations between living overweight or obesity and having high nutritional risk dietary patterns [10,11,12].

Evidence on the association between processed food consumption and body weight is conflicting. Adams and colleagues [13] found an inverse association between the consumption of “processed culinary ingredients” (sugars and oils) and higher body weight but did not identify an association between weight status and ultra-processed food product intake. In contrast, two cross-sectional studies [14,15] that analyzed pediatric population data from the United States and Brazil showed that a higher consumption of ultra-processed food products was associated with higher odds of being overweight and obese [16].

The divergence between nutritional status and food intake among Brazilian adolescents is noted to be present in clinical practice. Adolescents with eating and health practices, which represent a high nutritional risk, but are in an eutrophic nutritional status, are typical. Similarly, obese adolescents present health-promoting dietary practices. It is likely that this complex scenario of nutritional status and dietary intake is justified because adolescence is a period of significant physiological changes associated with puberty, which is impacted by environmental, genetic, nutritional, and psychosocial factors [17]. Additionally, obesity also has a multiple-factor mechanism as it does during puberty [18].

Intending to overcome the divergent associations between food intake and nutrition-al status, it was decided to analyze the food and nutritional profiles of Brazilian adolescents, which is from the combinations between dietary patterns and nutritional statuses. This measurement was based on a study [19] that developed a combination of nutritional status variables to assess the double burden of malnutrition. It is recognized that there are adolescents with distinct dietary and nutritional profiles, and these distinctions should be identified and analyzed for developing adequate dietary and nutritional care actions for this age group. The studies lack the construction of a more robust variable, and this must be analyzed in relation to the social determinants—the individuals’ behavioral and psychosocial health factors, the socioeconomic position and material circumstances of the individual and family, and economic context [20]. Considering all of the above, the present study analyzed the prevalence of dietary and nutritional profiles among Brazilian adolescents and their associations with social determinants of health.

## 2. Methods

### 2.1. Study Design and Data Source

This was a population-based survey from the “Sample 2” database of the 2015 PeNSE, which used a stratified and conglomerate complex sampling plan. This population study was conducted by the Brazilian Institute of Geography and Statistics (IBGE) and the Ministry of Health, with support from the Ministry of Education, and was approved by the National Research Ethics Committee with registration no. 1,006,467 [21]. The present work complies with the ethical precepts set forth in resolution no. 510 of 7 April 2016, of the National Health Council [22].

The “Sample 2” database of the 2015 PeNSE totaled 16,608 adolescents. These were students distributed from the sixth grade of elementary school to the third grade of high school in the morning, afternoon, and evening periods. All were enrolled and regularly attending public and private schools located in urban and rural areas throughout Brazil. With these data, it is possible to estimate the population parameters for Brazil and each macro-region (North, Northeast, Southeast, South, and Center–West). This study used a cluster sampling plan. The schools were divided into geographic extracts (primary sample unit) and then by cohort (secondary sample unit) through drawing. All students in the cohorts were invited to participate in the survey. The sample size in each extract considered the following parameters: approximate sampling error of 3% in absolute values for the estimate of 50%, confidence interval of 95%, and mean effect of the sampling plan equivalent to “3” in the first stage [21].

The data collection was conducted between April and September 2015 using an electronic self-administered questionnaire with the adolescents and a questionnaire directed to the school’s guardian. The questionnaire was developed based on the recommendations of the Global School-based Student Health Survey—GSHS, developed by the WHO. The measurement of weight and height occurred immediately after the completion of the questionnaire by the students, who were moved to a place other than the classroom that met the minimum requirements for anthropometry. Students with physical impairments that made it difficult to perform anthropometry did not have their measurements taken. Additionally, those who refused were not submitted to the procedure, as provided in the instruction manual of the survey. The weighing was conducted with portable electronic scales, and the weight was recorded in kilograms, considering the first decimal presented on the scale’s display. Height was measured using a portable stadiometer, fixed to a flat wall with the help of adhesive tape, and recorded in centimeters, considering the first decimal place. The data collection technicians were instructed to take two weight and height measurements and to repeat the third one if the previous ones were different; however, only one piece of information for each variable was recorded on the student’s smartphone. Further information about the research can be found in the PeNSE 2015 publication [21] and in articles already published by the research group [12,23].

The analyses described below used individual data from 16,556 adolescents including information on the weight, height, sex, and age available in the IBGE database. Regarding the total of 16,409 students available in the original database, the study lost no more than 147 individuals or 0.89%.

### 2.2. Study Variables

The eight dietary and nutritional profiles—the principal variable—resulted from combining the four categories of nutritional status with the two categories of dietary pattern: (1) lower nutritional risk dietary pattern and thinness; (2) higher nutritional risk dietary pattern and thinness; (3) lower nutritional risk dietary pattern and eutrophy; (4) higher nutritional risk dietary pattern and eutrophy; (5) lower nutritional risk dietary pattern and overweight; (6) higher nutritional risk dietary pattern and overweight; (7) lower nutritional risk dietary pattern and obesity; and (8) higher nutritional risk dietary pattern and obesity.

The identification of two eating patterns was performed by employing a cluster analysis with a non-hierarchical method (k-means), which separated the group of individuals according to an a priori definition of the number of clusters. The generated patterns were multivariate metrics constituted from the weekly frequency of seven food markers: beans, fruits, legumes, and vegetables (e.g., lettuce, pumpkin, broccoli, onion, carrot, carrot, chayote, cabbage, spinach, cucumber, tomato, etc. do not include potatoes and cassava), soft drinks, sweets (candies, chocolates, gum, or lollipops), ultra-processed salty snacks (hamburger, ham, mortadella, salami, sausage, frankfurter, instant noodles, snacks, crackers), and fried snacks (e.g., fried potatoes—not counting the potato chips in packaged snacks—or fried snacks such as chicken drumstick, fried kebab, fried pastry, “acarajé”, etc.). This resulted in a lower nutritional risk dietary pattern and a higher nutritional risk dietary pattern. All questions about the frequency of food intake were related to the number of times in the last seven days prior to the survey in which the adolescent had eaten foods from each of the seven groups, which may vary between zero (never) and seven times a week.

For the classification of the nutritional status, we considered the age range from 10 to 19 years, 11 months, and 29 days, according to the Brazilian Ministry of Health. The BMI was calculated by the formula [weight (kg)/height2 (m)]. The definition of thinness, eutrophy, overweight, and obesity among adolescents was based on the variables of sex, age, weight, and height. The z-score values for the indicator body mass index for age (BMI-I) were calculated in WHO AnthroPlus software [24]. Nutritional status variables were defined from the following cut-off points: marked thinness (BMI-I < Score-z −3), thinness (BMI-I ≥ Score-z −3 and < Score-z −2), eutrophy (BMI-I ≥ Score-z −2 and < Score-z +1), overweight (BMI-I ≥ Score-z +1 and < Score-z +2), obese (BMI-I ≥ Score-z +2 and < Score-z +3), and severely obese (BMI-I ≥ Score-z +3). The four categories of nutritional status used in the composition of the main variable were thinness (marked thinness + thinness), eutrophy, overweight, and obesity (obesity + severe obesity) [25,26].

Secondary variables were representative of the sociodemographic dimensions, and health behaviors were related to social determinants in health, as described in the following:The school’s sociopolitical and economic context: geographic macro-region (North, Northeast, Southeast, South, and Central–West).The school’s material circumstances: school situation (urban, rural), administrative dependence (public, private).The socioeconomic position and material circumstances of the individual and family: gender (male, female), age (10–14 years, 15–19 years), job (yes, no), maternal education level (uneducated, literate, primary school, high school, college, or did not know); number of residents in the household (≥5 residents, <5 residents).Individuals’ behavioral and psychosocial health factors: breakfast consumption, lunch or dinner consumption with parents or caregivers (regular ≥5 days, irregular <5 days), food consumption while watching TV or studying and having been to fast-food restaurants during the week before the survey (no, yes), practicing physical activity (<300 min/week, ≥300 min/week), have used formulas or medicines to lose or control weight without professional supervision (yes, no); have used formulas to gain weight or muscle mass without professional supervision (yes, no); have used laxatives to lose or control weight without professional supervision (yes, no).

### 2.3. Statistical Analyses

Descriptive analyses were performed by estimating the prevalence and confidence intervals for nutritional status, dietary patterns, and the eight dietary and nutritional profiles. The model estimation associated with the secondary variables and eight dietary and nutritional profiles was performed by correspondence analysis.

This is a statistical interdependence technique based on a graphic representation of the absolute frequencies of variables on a multidimensional map. It allows one to test the associations between categories of the principal and secondary variables. Variables that are in different quadrants and at close points on the map are associated, and variables that are not associated are represented in different quadrants [27].

## 3. Results

The prevalence of severe thinness (0.4%; 95%CI: 0.3–0.5), thinness (2.3%; 95%CI: 2.0–2.7), eutrophy (70.7%; 95%CI: 69.8–71.7), overweight (16.6%; 95%CI: 15.9–17.4), obesity (8.7%; 95%CI: 8.1–9.2), and severe obesity (1.3%; 95%CI: 1.1–1.6) among Brazilian adolescents. The highest prevalence of thinness was in individuals from the Northeast region (3.0%; 95%CI: 2.3–3.9); overweight in individuals with 10–14 years of age (19.0%; 95%CI: 18.0–20.1); obesity in adolescents aged 10–14 years (10.5%; 95%CI: 9.7–11.4), and from the South region (10.5%; 95%CI: 9.4–11.7) as well as severe obesity in the South region (2.1%; 95%CI: 1.6–2.7) (Table 1).

Two dietary patterns were identified based on the cluster analysis. The lower nutritional risk dietary pattern (61.6%; 95%CI: 60.6–62.6) was characterized by a higher intake of beans, fruits, vegetables, and greens during the week, and a lower of salty ultra-processed food products, soft drinks, sweets, and fried snacks. The higher nutritional risk dietary pattern was prevalent in 38.4% (95%CI: 37.4–39.4) of Brazilian adolescents and was characterized by a higher weekly intake of ultra-processed food products to the detriment of in natura or minimally processed foods (Figure 1).

The most prevalent food and nutritional profile among Brazilian adolescents was the lower nutritional risk dietary pattern and eutrophy (42.6%; 95%CI: 41.3–43.7). Relevant prevalence of complex profiles were found such as a higher nutritional risk dietary pattern and eutrophic pattern (28.1%) and lower nutritional risk dietary pattern with overweight (10.6%) and with obesity (6.8%) (Table 2).

Correspondence analysis associated these dietary and nutritional profiles with the secondary variables and identified four significant clusters arranged in different quadrants of the symmetrical graph and whose total represents a model that explained 92.83% of the data variability (Figure 2).
(1)The profiles “higher nutritional risk dietary pattern and thinness” and “higher nutritional risk dietary pattern and eutrophy” were associated with residents of the Northeast region, mothers with primary schools or college education, adolescents aged 15–19 years, used to consuming food, watching TV, or studying, had frequented fast food restaurants in the week before the survey, practiced less than 300 min of physical activity weekly, and having used formulas to gain weight or muscle mass (Figure 2).(2)The profiles “lower nutritional risk dietary pattern and thinness” and “lower nutritional risk dietary pattern and eutrophy” were associated with adolescents living in the North and Northeast regions, studying in public schools and rural areas, with five or more residents at home, with illiterate or literate mothers or where the adolescents did not know their mother’s education, male, who had not used formulas or laxative actions for weight loss, and with the regular consumption of breakfast and the regular consumption of lunch with parents or caregivers (Figure 2).(3)The profiles “lower nutritional risk dietary pattern and overweight ” or “lower nutritional risk dietary pattern and obesity ” were associated with adolescents living in the Central–West and Southeast, aged 10–14 years, with 300 min or more of physical activity per week, without frequenting fast food restaurants in the week before the survey, not eating while watching TV or studying, not having used formulas for gain weight or muscle mass, and regularly consumed lunch with parents or caregivers (Figure 2).(4)The profiles “higher nutritional risk dietary pattern and overweight ” and “higher nutritional risk dietary pattern and obesity ” were related to living in the South and Southeast regions, studying in urban and public schools, living in houses with less than five residents, having mothers with high school education, female, irregular consumption of breakfast, and irregular consumption of lunch or dinner with family members, and having used formulas or laxative actions to lose or control weight (Figure 2).

## 4. Discussion

The results indicate that different aspects of social, demographic, and behavioral aspects mark different lifestyles among adolescents with distinct dietary and nutritional profiles. It was observed that in Brazil, there was a high prevalence of adolescents with healthy dietary patterns and nutritional status. Adolescents with better dietary patterns (low nutritional risk) and without overweight (eutrophy or thinness) had worse socioeconomic position but lived in spaces with greater family participation (lower maternal education and regular sharing of meals with relatives), lived in less urbanized regions (North, Northeast, rural), and presented more appropriate eating and health behaviors.

The profiles characterized by a lower nutritional risk dietary pattern with thinness or eutrophy are therefore related to adolescents in situations of higher social vulnerability and worse material circumstances. This better dietary pattern is probably explained by the food culture of the territories located in the Brazilian North and Northeast, where the regional in natura or minimally processed foods are very present [28]. These large Brazilian regions are less urbanized areas where large groups are in poverty conditions that cause the highest prevalence of food insecurity and the lowest prevalence of overweight and a lower participation of ultra-processed food products in households [29].

The higher prevalence of adolescents experiencing hunger and social vulnerability in these territories [30] places them in conditions of lower access to ultra-processed food products (soft drinks, sweets, fried snacks, and salty ultra-processed foods) due to low purchasing power and limited financial resources for the regular purchase of these types of food. However, the food choices of these populations are highly affected by the context of the food system changes that have been causing the increased consumption of ultra-processed food products in these predominantly Amazonian territories. This transition has been happening, even though the results of the present study indicated a lower nutritional risk of food consumption among adolescents in 2015 and the results of another study demonstrated the maintenance of strong traditional food culture in the North region [31].

Therefore, these territories that present profiles of lower food and nutrition risk among adolescents should invest more in health promotion actions. These collective actions should aim to strengthen positive eating practices among adolescents with eutrophy and reverse the situations of thinness among those adolescents who experience such restrictions. For this, it is essential to develop actions to preserve healthy eating practices and to ensure the food security of this population has a better dietary and nutritional profile, ensuring a healthier adolescence and future.

In these territories, actions must be taken so that the positive situations promoting healthy and adequate eating (lower consumption of ultra-processed food products, regular consumption of breakfast, and the habit of having meals with the family) are not modified. In addition, it is necessary to assess other indicators related to food insecurity and integral health for the development of structuring actions to ensure the human right to adequate and healthy food. Variables such as access to food and the number of daily meals eaten by adolescents were not assessed by PeNSE, but are necessary for a better understanding of the material circumstances available to these groups.

Another interesting result was the presence of less educated mothers in these lower nutritional risk profiles. In Brazil, mothers with less education who live in these less urbanized territories are usually responsible for all activities within the household, preparing meals and providing adolescents with moments to share these meals with their families. This, together with the more traditional food culture, may explain the regular practice of consuming breakfast, lunch, and dinner with family members among this group. Certainly these practices explain the better dietary pattern of these dietary and nutritional profiles. It is noteworthy that another study identified less consumption of ultra-processed food products among adolescents from rural regions, children of mothers with less education, and worse socioeconomic conditions [32]. These issues should be further investigated in other studies for a better understanding of these contexts.

Studies have already pointed out the better quality of the diet among adolescents who regularly eat this first-morning meal [33], mainly because in Brazil, at this first moment of the day, they usually eat fruit together with energy sources (tapioca, couscous, slices of bread, or tubers), protein sources (eggs, meat, chicken), and calcium sources (milk and yoghurts) [28]. In addition, the results of studies with adolescents highlight the positive associations between a higher frequency of meals shared with the family and better markers of food, nutrition, and health in this age period [34,35].

Profiles with higher nutritional risk dietary patterns and thinness or eutrophy should be a priority in actions to promote healthy eating and health promotion. This is to reverse food practices related to the higher consumption of ultra-processed products such as frequenting fast-food restaurants and eating in the presence of screens. The variables associated with this profile indicate a group of adolescents with more independence in food choices because they are in family environments where the mothers have a higher level of school education. These Brazilian mothers usually spend most of their daily time working outside the home, making it difficult to provide nutritional and food guidance more effectively. Parental education strategies must be promoted on the need for an adolescent’s dietary and nutritional care shared with all adult family members. It is perceived that in Brazil, this is a function that is still very centralized with women [36,37].

The profile associated with adolescents from the Northeast region was compared with the results of [16] who studied North American low-income children and observed a higher intake of processed and ultra-processed food products but did not identify a significant relationship between this pattern of food consumption and nutritional status (weight status). It is likely that the coexistence of so many dietary and nutritional risk factors in this group can be explained by the relations of this adolescent in more urbanized territories within the Brazilian Northeast region, which make up food environments with a greater availability of ultra-processed food products and less healthy food [38]. Future studies with more specific territorial clippings (states or municipalities) may identify whether these higher nutritional risk dietary patterns are more prevalent in the more urbanized spaces of this large region.

Actions in the dimension of promoting healthy eating and health should be developed for this group of Brazilian adolescents, especially because they also present physical activity below those recommended by international health agencies. Certainly, this is a group that, in practice, does not implement activities to promote health and self-care because they do not consider it a necessary and immediate measure because they have excess weight (overweight or obesity). The actions to combat obesity need to be well-worked out when aimed at adolescents because a public health practice based only on campaigns issues health recommendations that may be understood by this age group as necessary only for those with established nutritional disorders. This profile is composed of a group of adolescents with appropriate anthropometric nutritional status, but by presenting these risky eating practices, may have or develop important metabolic dysfunctions in the medium and long-term [39].

In addition to these behaviors, adolescents in this profile who are older (15–19 years) do not seem to be a priority in food and nutrition education actions in Brazil. They are in high school, and this is a stage of schooling that the Brazilian educational system does not provide health actions, and when it does, they do not seem to be effective. Only with the approval of law 13.666/19 [40], which inserted food and nutrition education as a cross-cutting theme in the curriculum of Brazilian schools, is the timid inclusion of actions to promote healthy eating behaviors in educational spaces that serve the public of this age group beginning to be perceived [41].

Among the complex eating patterns, this study also identified the profiles of better dietary pattern and overweight or obesity that were associated with important positive health practices such as eating with the family and performing physical activities according to the recommendations of international organizations (more than 300 min per week). Such profiles were related to younger adolescents and the non-existence of eating behaviors with a negative effect on health (frequenting fast food, eating in the presence of screens, using formulas for weight or muscle mass gain). In the context of collective health actions, adolescents with these profiles should be the target of guidance and actions to strengthen their health-promoting behaviors in the environments where they live such as families, schools, and health services.

This group is an important group due to the nutritional problems already established (overweight and obesity), but actions can be guided by the proposals of health practices (quality of sleep, regular and diversified physical activity, periodic assessment of metabolic risk) and the incorporation of new health-promoting eating behaviors (regular consumption of breakfast and better quality of food consumed). All of this may help to reverse the problems of overweight and obesity, considering that because they are in the first half of adolescence (10–14 years old), they are mostly at the time of puberty transformations that cause metabolic, height, and body composition changes [42,43] and should be accompanied by multi professional teams qualified for the treatment of obesity in adolescents [18,44].

Moreover, the existence of these dietary and nutritional profiles in the Central–West and Southeast may be related to the “caipira” food culture present in these territories, which also values culinary practices, fresh food, and the consumption of fruit, vegetables, and legumes, but with a greater presence of sugars and the use of fats in culinary preparations [28,45].

The adolescents’ dietary and nutrition profiles with the highest negative impact on health are those composed of higher nutritional risk dietary patterns coexisting with overweight and obesity. The estimation of these profiles suggests a relationship with the female gender. This result highlights girls as a priority for public actions to prevent overweight and other health issues such as the risk behavior for eating disorders. Identifying this profile in populations is critical because it refers to the coexistence of multiple nutritional, dietary, and behavioral risk factors. In Brazil, this is associated with a group of more urbanized territories (Southeast and South) with greater socioeconomic privileges (private schools), but seem to lack more connections and guidance in their family [37,46] and school environments [47]. Further studies need to be conducted to better understand this relationship between gender and social position, as they are known to interact in determining other adolescent health risk behaviors such as smoking, violence, unhappiness, and mental illness [48].

Additionally, this profile, which is inserted in private schools, can have their dietary and nutritional issues accommodated and worked in this educational environment as a priority health action. Adolescents from Brazilian private schools have worse markers of diet and nutrition, possibly because they do not have regular access to food and nutrition education actions that are regulated by national guidelines and developed in public schools but are not common in private schools [49], or even because of the higher availability of ultra-processed food products in these spaces in which adolescents have more purchasing power for acquisition and consumption [28]. Studies on the school food environment can help to better understand the determinants of overweight and obesity. Studies developed in Brazil have already pointed out the role of the ultra-processed food products available and marketed inside and around schools and the higher prevalence of overweight and obesity [50].

These two dietary and nutritional profiles that combine a higher nutritional risk dietary pattern with overweight and obesity among adolescents are the priority realities for health actions. The metabolic risk resulting from overweight and obesity, together with the high consumption of ultra-processed food products (such as sugary drinks and ultra-processed snacks) and the insufficient consumption of fresh and minimally processed foods (such as fruits, vegetables, and beans) is recognized [51].

Ultra-processed food products have a negative impact on health not only due to their nutritional profile (high in sodium, simple sugar, and total fat; low in fiber, vitamins, and minerals) but also due to mechanisms associated with foods that produce excessive consumption such as the intensification of sensory attributes. Evidence indicates that these products are gradually becoming predominant in the global food system in Brazil, and the growth in the amount of ultra-processed food products in the diet of the population has been observed in recent decades [13,15,52].

Adolescents experience the deleterious effects of the excessive consumption of these ultra-processed food products. Evidence from neuroscience studies highlights the risk of dietary patterns based on these types of foods, even on healthy brain development among adolescents and the dysregulation of decision-making, reward, and self-regulation pathways [53,54,55]. Such studies highlight the deleterious effects of frequent consumption of ultra-processed food products such as sweets, salty ultra-processed foods, and soft drinks, which should be seen not only as sources of calories and nutrients (fats, simple sugars, sodium) that in excess add a higher cardiometabolic risk, but now as a source of deregulatory effects on adolescent eating behaviors. Further studies should evaluate the pathways of action of a dietary pattern based on these health risk foods in the development and maintenance of obesity conditions.

These relationships between the consumption of ultra-processed food products and the neurological dysregulation of adolescent eating behavior may explain part of the association between the eating patterns of higher nutritional risk and overweight or obesity with risk behaviors for eating disorders (use of laxative actions and formulas for weight loss without being monitored by doctors or nutritionists) among adolescents in Brazil. Another explanation for this result is the expressive existence of aesthetic pressures suffered by girls regarding body shapes and standards. This causes negative impacts on health [56,57], and the possible forms of the management of overweight and obesity conditions based on the stigma of weight may worsen the health and obesity conditions and can trigger prodromal behaviors of eating disorders [44,58].

This study has limitations in relation to the PeNSE survey such as not considering students who did not attend school and evaluating food intake from seven food group markers. It is emphasized that the assessment of food intake from more robust elements (food diary or 24 h food collection) would allow for a better understanding of the total dietary variation. However, even with these weaknesses, the study has significant validity for its territorial coverage and for its pioneering combination of dietary patterns and different nutritional statuses for the understanding of how the resulting profiles are associated in the different life contexts of Brazilian adolescents.

## 5. Conclusions

Brazilian adolescents have different dietary and nutritional profiles that are associated with very distinct eating behaviors, health practices, and living conditions. The eight profiles analyzed appeared in the model, distributed in four groups marked by specific combinations that are characterized by sociopolitical and economic contexts, material family and school circumstances, and the behavioral and psychosocial health factors of individuals. Thus, emphasizing the dietary and nutritional profiles of adolescents results from the coexistence and interactions between the individual choices and different contexts of family and community environments. All of this points to the social determination of these health problems among adolescents in Brazil.

These results on the food and nutrition profiles and their coexistence with the different lifestyles of Brazilian adolescents contribute to the process of the food and nutrition surveillance of this population. The identification of these profiles and their distribution in the public allows for a better diagnosis of the food and nutritional realities in adolescence and the programming of efficient, effective, and more resolutive actions in the dimensions of the promotion, prevention, and management or treatment of the food and nutritional challenges faced by this age group.

## Figures and Tables

**Figure 1 nutrients-14-04233-f001:**
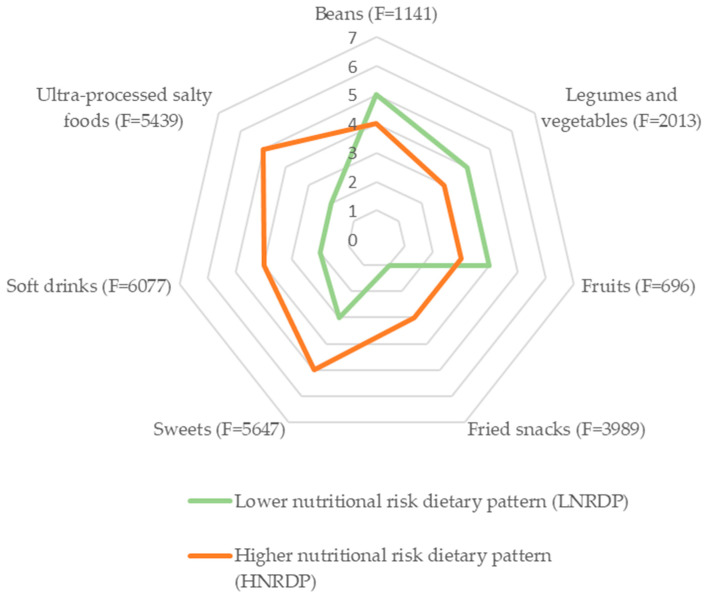
Average weekly food intake frequency according to the Brazilian school adolescents’ dietary pattern and F. statistic, according to cluster analysis. National School Health Survey (PeNSE), 2015. Note: Cluster I (lower nutritional risk dietary pattern) = 10,257 individuals (61.6%; 95%CI: 60.6–62.6) and Cluster II (higher nutritional risk dietary pattern) = 6153 individuals (38.4%; 95%CI: 37.4–39.4). *p* value < 0.001 for the Analysis of Variance (ANOVA) of the cluster analysis.

**Figure 2 nutrients-14-04233-f002:**
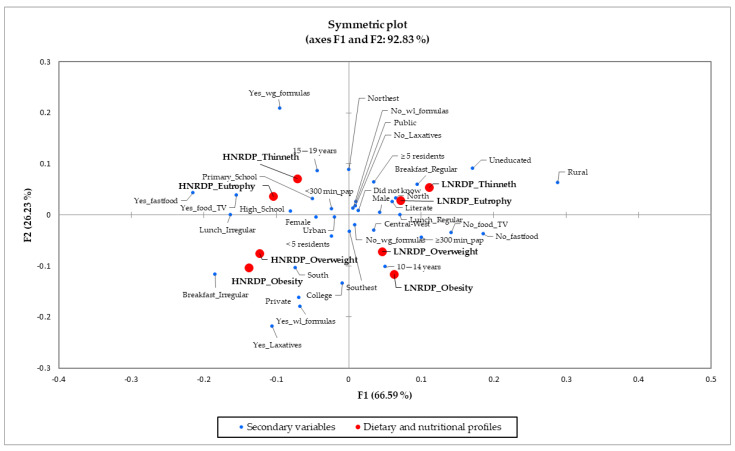
Correspondence analysis of Brazilian adolescents’ dietary and nutritional profiles and socio-economic and behavioral variables. National School Health Survey, 2015. Note: HNRDP_Thinneth: Higher nutritional risk dietary pattern and thinness; LNRDP_Thinneth: Lower nutritional risk dietary pattern and thinness; HNRDP_Eutrophy: Higher nutritional risk dietary pattern and eutrophy; LNRDP_Eutrophy: Lower nutritional risk dietary pattern and eutrophy; HNRDP_Overweight: Higher nutritional risk dietary pattern and overweight; LNRDP_Overweight: Lower nutritional risk dietary pattern and overweight; HNRDP_Obesity: Higher nutritional risk dietary pattern and obesity; LNRDP_Obesity: Lower nutritional risk dietary pattern and obesity; North, Northeast, Southeast, South, and Central-West: Geographic macro-region; Urban, Rural: School situation; Public, Private: School administrative dependence; Male, Female: Gender; 10–14 years, 15–19 years: Age; uneducated, literate, primary school, high school, college, or did not know: Maternal education level; ≥5 residents, <5 residents: Number of residents in the household; Breakfast_Irregular: Breakfast consumption < 5 days/week; Breakfast_Regular: Breakfast consumption ≥ 5 days/week; Lunch_Irregular: Lunch or dinner consumption with parents or caregivers < 5 days/week; Lunch_Regular: Lunch or dinner consumption with parents or caregivers ≥ 5 days/week; No_food_TV: No food consumption while watching TV or studying, Yes_food_TV: Food consumption while watching TV or studying having; No_fastfood: Had not been to fast-food restaurants in the week before the survey; Yes_fastfood: Had been to fast-food restaurants in the week before the survey; <300 min_pap: <300 min/week practicing physical activity; ≥300 min_pap: ≥300 min/week practicing physical activity; Yes_Laxatives: Had used laxative actions for weight loss; No_Laxatives: Had not used laxative actions for weight loss; Yes_wg_formulas: Had used formulas for gain weight or muscle mass; No_wg_formulas: Had not used formulas for gain weight or muscle mass; Yes_wl_formulas: Had used formulas to lose or control weight; No_wl_formulas: Had not used formulas to lose or control weight.

**Table 1 nutrients-14-04233-t001:** Nutritional status prevalence among Brazilian adolescents, according to the BMI-age indicator and socio-demographic variables. National School Health Survey (PeNSE), 2015.

Variables	Severe Thinness		Thinness		Eutrophy		Overweight		Obesity		Severe Obesity	
	%	CI 95%	%	CI 95%	%	CI 95%	%	CI 95%	%	CI 95%	%	CI 95%
Brazil	0.4	0.3–0.5	2.3	2.0–2.7	70.7	69.8–71.7	16.6	15.9–17.4	8.7	8.1–9.2	1.3	1.1–1.6
Geographic macro-region												
North	0.3	0.1–0.6	2.3	1.7–3.1	72.4	70.4–74.2	16.0	14.6–17.6	8.3	7.2–9.5	0.8	0.5–1.3
Northeast	0.4	0.2–0.7	3.0	2.3–3.9	73.7	71.8–75.5	14.8	13.3–16.4	7.2	6.2–8.3	1.0	0.7–1.4
Southeast	0.4	0.2–0.8	2.0	1.5–2.6	69.6	67.8–71.4	17.4	16.0–18.9	9.1	8.1–10.3	1.4	1.0–1.9
South	0.3	0.1–0.6	1.8	1.3–2.5	66.8	65.0–68.6	18.5	17.0–20.0	10.5	9.4–11.7	2.1	1.6–2.7
Central–West	0.4	0.2–0.7	2.0	1.5–2.6	69.9	68.0–71.7	17.4	15.9–18.9	9.2	8.1–10.4	1.3	0.9–1.9
Gender												
Male	0.4	0.3–0.7	2.6	2.1–3.1	70.9	69.5–72.2	15.7	14.6–16.8	9.0	8.2–9.8	1.5	1.2–1.9
Female	0.3	0.2–0.6	2.0	1.6–2.5	70.6	69.2–71.9	17.6	16.5–18.8	8.3	7.6–9.2	1.1	0.8–1.4
Age												
10–14 years	0.4	0.3–0.7	2.1	1.7–2.5	66.6	65.3–67.9	19.0	18.0–20.1	10.5	9.7–11.4	1.3	1.1–1.7
15–19 years	0.3	0.2–0.6	2.5	2.0–3.0	74.3	72.9–75.7	14.6	13.5–15.8	7.1	6.3–7.9	1.3	1.0–1.6

Note: severe thinness (BMI-I < Score-z −3), thinness (BMI-I ≥ Score-z −3 and < Score-z −2), eutrophy (BMI-I ≥ Score-z −2 and < Score-z +1), overweight (BMI-I ≥ Score-z +1 and < Score-z +2), obesity (BMI-I ≥ Score-z +2 and < Score-z +3), and severe obesity (BMI-I ≥ Score-z +3).

**Table 2 nutrients-14-04233-t002:** Prevalence of dietary and nutritional profiles among Brazilian adolescents (*n* = 16,409). National School Health Survey (PeNSE), 2015.

Dietary and Nutritional Profiles	N	%	CI 95%
Lower nutritional risk dietary pattern and thinness	234	1.5	1.30–1.80
Lower nutritional risk dietary pattern and eutrophy	6833	42.6	41.6–43.7
Lower nutritional risk dietary pattern and overweight	1915	10.6	10.0–11.3
Lower nutritional risk dietary pattern and obesity	1274	6.8	6.3–7.4
Higher nutritional risk dietary pattern and thinness	158	1.1	0.9–1.4
Higher nutritional risk dietary pattern and eutrophy	4396	28.1	27.1–29.0
Higher nutritional risk dietary pattern and overweight	1055	6.0	5.6–6.6
Higher nutritional risk dietary pattern and obesity	544	3.2	2.8–3.5

## Data Availability

Not applicable.

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
