# Peer review of "Dietary and Nutritional Profiles among Brazilian Adolescents"

_nutrients, 2022, doi:10.3390/nu14204233_

Round 1
Reviewer 1 Report
In the present study, Vale and colleagues evaluated the prevalence of dietary and nutritional patterns among Brazilian adolescents, and their associations with some social determinants of health. Based on their results, the authors state that Brazilian adolescents with distinct dietary profiles have different demographic, social and behavioral characteristics.
Various previous articles on the same topic are available in literature, therefore the strengths of this study are essentially the specific ethnic group investigated, the vastness of coverage and, in part, the statistical approach used. The manuscript is easy to read even if excessively verbose especially in the discussion which, in my opinion, should be shortened.
Major comments
Page 2, line 98. The authors claim to have used a self-administered electronic questionnaire but did not provide further details on data acquisition, except by referring to the previous article of 2019 (ref. no. 19) which, however, mentions generically the use of smartphones without specifying how information bias has been minimized. The authors should better clarify this point.
Page 3, methods. Why the analysis has not been carried out by stratifying according to ethnic groups? Adolescent of indigenous and black ancestry are not mentioned among the subgroups of adolescents studied.
Page 3, lines 119-121. The number of seven food markers was perhaps insufficient to cover the adolescent food profile necessary for a cluster analysis. Don’t the authors believe this contributed to capture only a small fraction of the total dietary variation?
Page 4, lines 172-175. Why was the prevalence of overweight/obesity so low? A prevalence of over 32% has previously been reported among adolescents in Brazil and, in general, in Latin America.
Page 4, Table 1. The prevalence of overweight/obesity was higher in the younger age groups and decreased in the older ones, but this was not adequately discussed further. Do the authors have any explanation of this finding? Is it possible to hypothesize a positive role of the school in the nutritional education of these subjects during their education course?
Page 7. The biplot has a low resolution and it is difficult to read the variable names well.
Page 8. The discussion, in addition to being quite verbose, contains some oversimplistic explanations such as that in rural areas, economic conditions and the low education of mothers, guarantee a better food quality for adolescents. Too often, the role of ultra-processed foods is blamed without making a proper distinction between “industrially processed foods” and “home processed foods” (the latter being often minimally processed) which do not have the same impact on health. Furthermore, an important point is that the food choices of adolescents, rather than arising from personal motivation, are largely conditioned by the availability of food at the local level, if not by the will and values ​​of parents and peers, which makes it important to take these aspects into account; a school-age teenager often has the financial resources to purchase snack foods, even if they are definitely not health promoting. The discussion, in addition to being shortened, requires in my opinion that these points be clarified.
Author Response
Response to Reviewer 1 Comments
Dear reviewer,
we thank you for your consideration in reviewing our scientific paper. The language of the text was reviewed by an English-language specialist and some points were added following advice from other reviewers. Your suggestions have been added to our text as per the following answers:
Point 1: Page 2, line 98. The authors claim to have used a self-administered electronic questionnaire but did not provide further details on data acquisition, except by referring to the previous article of 2019 (ref. no. 19) which, however, mentions generically the use of smartphones without specifying how information bias has been minimized. The authors should better clarify this point.
Response 1: We have added more information about the sampling and data collection process to explain important actions to reduce information bias.
Point 2: Page 3, methods. Why the analysis has not been carried out by stratifying according to ethnic groups? Adolescents of indigenous and black ancestry are not mentioned among the subgroups of adolescents studied
Response 2: It was not performed by stratification according to ethnic groups in the present study, because results of previous studies showed no associations between nutritional status and ethnicity, even in multivariate analyses performed with the same database as in the present study. The following is the reference of the article on which our decision was based: Vale, D.; Andrade, M.E.d.C.; Dantas, N.M.; Bezerra, R.A.; Lyra, C.d.O.; Oliveira, A.G.R.d.C. Social Determinants of Obesity and Stunting among Brazilian Adolescents: A Multilevel Analysis. Nutrients 2022, 14, 2334. https://doi.org/10.3390/nu14112334.
Point 3: Page 3, lines 119-121. The number of seven food markers was perhaps insufficient to cover the adolescent food profile necessary for cluster analysis. Don’t the authors believe this contributed to capturing only a small fraction of the total dietary variation?
Response 3: We agree with the reviewer on the limitation of using only seven dietary markers to understand adolescents' dietary profile. We do not believe that this invalidates the development of the study considering the plausibility of our results, in addition to the sample design and sample size having met the recommendations for conducting a cluster analysis. Therefore, we considered his observation and added a passage highlighting it as a limitation at the end of the discussion.
Point 4: Page 4, lines 172-175. Why was the prevalence of overweight/obesity so low? A prevalence of over 32% has previously been reported among adolescents in Brazil and, in general, in Latin America.
Response 4: We checked the prevalence in the database and the prevalence estimates are correct. Certainly, other articles may have found higher prevalences when adding up the frequencies of the overweight, obese, and severely obese categories. Combining these three categories in our study, we found an approximate prevalence of 26.6 among Brazilian adolescents. In the South of Brazil, a more urbanized region, this sum is approximately 31.1%.
Point 5: Page 4, Table 1. The prevalence of overweight/obesity was higher in the younger age groups and decreased in the older ones, but this was not adequately discussed further. Do the authors have any explanation for this finding? Is it possible to hypothesize a positive role of the school in the nutritional education of these subjects during their education course?
Response 5: Since the central aim of our study was to assess dietary and nutritional profiles and associated factors, we decided to discuss these profiles. We did not discuss the issue of obesity reduction with increasing age because we consider this information to be secondary in this scientific article. We believe that a more plausible explanation is the body changes due to puberty and the higher number of adolescent males in the sample. We thank you for your observation and for sharing the hypothesis about the relationship between the decrease in obesity and the actions of food and nutrition education. After discussing with our research group, we will try to test this hypothesis in our database. Due to the length of our discussion that has already been cut after your suggestions, we have decided not to add your suggestion to expose such a hypothesis in this study. Should you find it essential to include it, we are willing to plan to add it to the discussion after a new request.
Point 6: Page 7. The biplot has a low resolution and it is difficult to read the variable names well.
Response 6: We have replaced the previous plot with a new one with better resolution.
Point 7: Page 8. The discussion, in addition to being quite verbose, contains some oversimplistic explanations such as that in rural areas, economic conditions and the low education of mothers, guarantee a better food quality for adolescents. Too often, the role of ultra-processed foods is blamed without making a proper distinction between “industrially processed foods” and “home processed foods” (the latter being often minimally processed) which do not have the same impact on health. Furthermore, an important point is that the food choices of adolescents, rather than arising from personal motivation, are largely conditioned by the availability of food at the local level, if not by the will and values ​​of parents and peers, which makes it important to take these aspects into account; a school-age teenager often has the financial resources to purchase snack foods, even if they are definitely not health promoting. The discussion, in addition to being shortened, requires in my opinion that these points be clarified.
Response 7: We added a reference that supports the statement about dietary patterns marked by lower consumption of ultra-processed foods among adolescents from rural regions, who live in families with worse socioeconomic conditions and with mothers with less education: “32. Silva JB, Elias BC, Warkentin S, Mais LA, Konstantyner T. Factors associated with the consumption of ultra-processed food by Brazilian adolescents: National Survey of School Health, 2015. Revista Paulista de Pediatria 2022, 40, e2020362. https://doi.org/10.1590/1984-0462/2022/40/2020362”. We added in the article the indication of the need for studies that better assess this context since we did not identify other studies to better explain this reality that is perceived in our territory. Based on this questioning, our research group plans to carry out our analyses by stratifying better the family contexts of adolescents.
We added in the methodology the examples of ultra-processed foods contained in the food marker questions asked to the students at the time of data collection. In addition, we replaced the term "ultra-processed foods" with "ultra-processed food products" throughout the text. With this, we believe in clarifying that the ultra-processed foods described in the article are industrial formulations that have proven negative effects on health. These are different from the culinary preparations highlighted in the reviewer's comment.
We have adapted parts of the text (discussion and conclusions) to highlight that the dietary and nutritional profiles among adolescents do not stem solely from individual choices. In addition, some paragraphs of the discussion have been adapted following the guidance to abbreviate this part of the text.
Best regards,
Reviewer 2 Report
Some small suggestions:
Introduction
- The last two paragraphs of this section could be shortened. It is necessary to say that the studies lack the construction of a more robust variable and that this must be analyzed in relation to social determinants. That seems enough to me.
Results
- Table1 should mention as a note the following description: marked thinness (BMI-I < Score-z -3), thinness (BMI-I ≥ 132 Score-z -3 and < Score-z -2), eutrophy (BMI-I ≥ Score-z -2 and < Score-z +1), overweight 133 (BMI-I ≥ Score-z +1 and < Score-z +2), obese (BMI-I ≥ Score-z +2 and < Score-z +3), and 134 severely obese (BMI-I ≥ Score-z +3).
Author Response
Response to Reviewer 2 Comments
Dear reviewer,
we thank you for your consideration in reviewing our scientific paper. The language of the text was reviewed by an English-language specialist and some points were added following advice from other reviewers. Your suggestions have been added to our text as per the following answers:
Point 1: Introduction - The last two paragraphs of this section could be shortened. It is necessary to say that the studies lack the construction of a more robust variable and that this must be analyzed in relation to social determinants. That seems enough to me.
Response 1: Reductions have been made in the last two paragraphs of the introduction. We did not make substantial changes because this would cause the removal of two important theoretical references from our study.
Point 2: Results - Table1 should mention as a note the following description: marked thinness (BMI-I < Score-z -3), thinness (BMI-I ≥ 132 Score-z -3 and < Score-z -2), eutrophy (BMI-I ≥ Score-z -2 and < Score-z +1), overweight (BMI-I ≥ Score-z +1 and < Score-z +2), obese (BMI-I ≥ Score-z +2 and < Score-z +3), and severely obese (BMI-I ≥ Score-z +3).
Response 2: The note was added to Table 1.
Best regards,
Reviewer 3 Report
In my opinion, the manuscript was prepared well, so I suggest accepting it in actual form.
* The Introduction Section explains the design of the study. The Authors well justify the research topic.
* The study was carried out without methodological errors.
* The Descriptions of the results were correct.
* The presented figures and table were prepared precisely and also legible.
* The Discussion Section includes the accurate reference of the results obtained to the
In the study authors present dietary and nutritional profiles among Brazilian adolescents; the prevalence of these profiles, and correlations of these profiles with some social determinants of health. The question addressed in the paper, and the whole paper is relevant and interesting. The topic is original and up-to-date. The paper presents dietary and nutritional profiles specific to Brazil, which not have been published previously. These profiles are characterized by sociopolitical and economic contexts and material family, and school circumstances, which also have not been shown so far. The paper is well written and clear to read, with scientific and proper language. The conclusions are consistent with the evidence and arguments presented. Conclusions address the main question of the paper and give answers to the study's aim.
Author Response
Dear reviewer,
we thank you for your consideration in reviewing our scientific paper. The language of the text was reviewed by an English-language specialist and some points were added following advice from other reviewers.
Best regards,
Round 2
Reviewer 1 Report
The authors have satisfactorily replied to my previous remarks by modifying the text accordingly. When it was not possible to address the requests, they motivated it in an appropriate manner. In particular, I appreciated the new details provided on the use of cluster sampling in the method section. As far as I can judge, the authors have also improved the text language. However I observe that at line 116 of the revised manuscript the term “available” was perhaps more appropriate, because I think “disposable” has another meaning in English. Figure 2 is still a bit difficult to read but has improved compared to the previous version. I have nothing else to ask.
Author Response
Dear reviewer,
We thank you for your careful review of our scientific paper. We have changed the term "disposable" to "available" in the text. We agree that this term was misapplied. We have tried to improve the resolution of figure 2 further and adjusted the layout of the text within the figure to enhance readability and understanding. We hope that this plot has improved. Reading this figure may be initially complicated due to the number of variables that we had to add to the model of dietary and nutrient profiles among Brazilian adolescents.
With best regards,